# Incorporation of Hydroxyapatite into Glass Ionomer Cement (GIC) Formulated Based on Alumino-Silicate-Fluoride Glass Ceramics from Waste Materials

**DOI:** 10.3390/ma14040954

**Published:** 2021-02-18

**Authors:** Wan Nurshamimi Wan Jusoh, Khamirul Amin Matori, Mohd Hafiz Mohd Zaid, Norhazlin Zainuddin, Mohammad Zulhasif Ahmad Khiri, Nadia Asyikin Abdul Rahman, Rohaniah Abdul Jalil, Esra Kul

**Affiliations:** 1Department of Physics, Faculty of Science, Universiti Putra Malaysia, UPM Serdang, Selangor 43400, Malaysia; mimieywanjusoh@gmail.com (W.N.W.J.); mhmzaid@upm.edu.my (M.H.M.Z.); nadiaxrahman84@gmail.com (N.A.A.R.); niajalil20@gmail.com (R.A.J.); 2Material Synthesis and Characterization Laboratory, Institute of Advanced Technology, Universiti Putra Malaysia, UPM Serdang, Selangor 43400, Malaysia; mzulhasif@gmail.com; 3Department of Chemistry, Faculty of Science, Universiti Putra Malaysia, UPM Serdang, Selangor 43400, Malaysia; norhazlin@upm.edu.my; 4Department of Prosthodontics, Faculty of Dentistry, Ataturk University, 25030 Erzurum, Turkey; esra.kul@atauni.edu.tr

**Keywords:** ASF glass ceramics, clam shell, glass ionomer cement, hydroxyapatite, soda lime silica glass

## Abstract

Glass ionomer cement (GIC) is a well-known restorative material applied in dentistry. The present work aims to study the effect of hydroxyapatite (HA) addition into GIC based on physical, mechanical and structural properties. The utilization of waste materials namely clam shell (CS) and soda lime silica (SLS) glass as replacements for the respective CaO and SiO_2_ sources in the fabrication of alumino-silicate-fluoride (ASF) glass ceramics powder. GIC was formulated based on ASF glass ceramics, polyacrylic acid (PAA) and deionized water, while 1 wt.% of HA powder was added to enhance the properties of the cement samples. The cement samples were subjected to four different ageing times before being analyzed. In this study, the addition of HA caused an increment in density and compressive strength results along with ageing time. Besides, X-ray Diffraction (XRD) revealed the formation of fluorohydroxyapatite (FHA) phase in HA-added GIC samples and it was confirmed by Fourier Transform Infrared (FTIR) analysis which detected OH‒F vibration mode. In addition, needle-like and agglomeration of spherical shapes owned by apatite crystals were observed from Field Emission Scanning Electron Microscopy (FESEM). Based on Energy Dispersive X-ray (EDX) analysis, the detection of chemical elements in the cement samples were originated from chemical compounds used in the preparation of glass ceramics powder and also the polyacid utilized in initiating the reaction of GIC.

## 1. Introduction

A form of dental restoration and luting material, called glass ionomer cement (GIC), is widely used in dental application. In the early of 70′s, Alan Wilson and Brian Kent introduced GIC as water-based cement derived from silicate-based glass and polycarboxylate salt [1]. This water-based substance forms essentially when there is a reaction between aqueous polyacrylic acid solution and fluoro-aluminosilicate glass powder [1,2,3].

GIC has desirable properties which encourage their clinical uses in dentistry such as it provides a direct bonding to tooth structure [4,5,6], exhibit a good biocompatibility and has the ability to release fluoride which is important as anticariogenic agent in preventing the occurrence of tooth decay [7]. Furthermore, the higher stability of GIC when involved in aqueous environment and also the results of low toxicity when applied to tooth are the advantages of GIC in dental application. Despite having good properties, there is some limitation that limits GIC from extensive use in dentistry as filling and restorative material. GIC has been shown to have weak mechanical and physical properties including low fracture strength and hardness, reduced wear resistance and opaqueness [8]. Besides, GIC is very brittle and this proved the low physical strength of GIC to be used as dental material.

Therefore, there are many efforts that have been focused on the modification of GIC so that the physical and mechanical properties of GIC can be enhanced. One of the ways is by incorporating hydroxyapatite into the GIC composition. Hydroxyapatite (HA) is an inorganic compound with a constitution equivalent to natural bone mineral and is well known as an apatite group (a group of mineral phosphate). HA is represented by the chemical formula of Ca_10_(PO_4_)_6_(OH)_2_, differ with other apatite compounds by containing hydroxyl end-member of the apatite group [4,5]. HA present as a major component of mineral in bone and teeth with calcium-to-phosphate ratio of 1.67 (Ca/P = 1.67) and hexagonal crystal structure [4,5,8]. According to Ramsden and co-workers, HA has osteoconductive and bioactive properties which make it favorable in orthopedics and dental application [9].

Alumino-silicate-fluoride (ASF) glass ceramics composition used in the formulation of GIC is believed to have good properties and favorable for dental materials [10,11,12]. The design of ASF glass ceramics composition is based on the 45S5 bioactive glass system used by Larry L. Hench which implicated a combination of glass network formers, modifiers and intermediates oxides [13,14]. The use of waste materials like clam shell (CS) and soda lime silica (SLS) glass as replacement of silicate and calcite sources, respectively in the synthesis of glass ceramics can save the production cost and reduce the environmental issues regarding the disposal problems. According to previous literatures, the general composition of SLS glass consists of 70–75 wt.% of SiO_2_, 12–16 wt.% of Na_2_O and 10–15 wt.% of CaO [15,16], meanwhile CS contains more than 98% CaC which encouraged it uses as biomaterial [17].

The key purpose of this work is to investigate the influence of HA addition on the composition of glass ceramics in GIC formulation based on physical, mechanical and structural properties. Although some work on the addition of HA into GIC has been performed, a deeper study can be gained especially involving the participation of ASF glass ceramics as the base silicate powder in the formulation of GIC. ASF glass ceramics composition with the existence of fluorapatite (FA) had been studied by several researchers [10,11,12] and the involvement of FA crystal phase in the GIC formulation is crucial since it can give effect to the properties of the resulting cement. Moreover, the utilization of waste materials which are CS and SLS glass in the fabrication of ASF glass ceramics and subsequently formulated into GIC can attract a great interest to researchers since there is limited studies on the use of waste materials in the GIC formulation. However, the GIC formulated based on ASF glass ceramics as base silicate powder was observed to produce inadequate mechanical strength which restricted their uses as dental material. Thus, the study on the addition of HA into the GIC formulation can help in improving the properties of resulting GIC. Besides, the effect of ageing time of the cement samples was also investigated in this study.

## 2. Materials and Experimental Procedure

### 2.1. Materials

Raw materials used as sources of CaO and SiO_2_ were derived from CS and SLS glass, respectively. *Anadara granosa* sp. CS were collected from the beachside of Pantai Cahaya Bulan, Kelantan, Malaysia, while SLS glass which was ‘Jalen’ soy sauce brand obtained from restaurants near to Universiti Putra Malaysia, Selangor, Malaysia.

The glass ceramics component of the cement involved the mixing of raw materials and chemical components such as CaF_2_ (R&M Chemicals, Semenyih, Selangor, Malaysia; 99.95%), P_2_O_5_ (Alfa Aesar, a Johnson Matthey Company, Lancashire, UK; 99.99%) and Al_2_O_3_ (Alfa Aesar, a Johnson Matthey Company, UK; 99.5%). Meanwhile, HA powder used was reagent grade powder (Sigma Aldrich, a Merck KGaA Company, Darmstadt, Germany; ≥90%) with CAS No. 1306-06-5 and molecular weight (M_W_) ~502.31. The participation of dried powder of polyacrylic acid (PAA) with M_W_ ~30,000 obtained from Advance Healthcare Ltd., Willenhall, UK and deionized water as a medium of reaction was crucial in the cement formation.

### 2.2. Preparation of CS and SLS Glass Powder

Preparation of raw materials was started by the cleaning process of collected CS and SLS glass in order to remove dirt from the materials. For CS, calcium oxide (CaO) was obtained from the calcination process of CS at 900 °C for 3 h as explained in several studies [10,11,12]. In this process, carbon dioxide (CO_2_) in the form of gas was removed from aragonite and calcite (CaCO_3_) which were the majority components of the CS. After that, both CS and SLS glass went through the same process, whereby they were crushed, plunged and sieved before getting ≤45 µm sized CS and SLS glass powder.

### 2.3. Synthesis of ASF Glass Ceramics Composition

ASF composition was fabricated by mixing 25 wt.% SLS, 20 wt.% CS, 20 wt.% P_2_O_5_, 20 wt.% Al_2_O_3_ and 15 wt.% CaF_2_ with 30 g of the total weight composition. After the five components were mixed homogeneously in alumina crucible, they were subjected to a melting process at 1500 °C in the furnace for 4 h. Next, the quenching process was done by pouring the molten glass into water, thus producing frits. Subsequently, the obtained frits were manually crushed by using plunger and grinded by using mortar and pestle, which then formed ≤45 µm-sized powder of ASF glass ceramics.

### 2.4. Formulation of Control and Modified GIC Samples

GIC was formulated by ASF glass ceramics, PAA and deionized water with ratio of 3:1:1 (glass ceramics:PAA:water). The setting reaction of cement required the combining of the three components on the glass plate for approximately 60 s. Right after 60 s, the cement was filled and pressed into stainless-steel mold (6 mm × 4 mm) to produce cylindrical shaped pellets and then dried at 37 °C for an hour. Next, the cement went through ageing process for 1, 7, 14 and 21 days in deionized water before characterized. In this study, a total of 30 pellets of GIC and 30 pellets of GIC1 samples were formed. GIC sample represented the GIC without HA addition while GIC1 sample represented GIC with the addition of 1 wt.% of HA powder into GIC. The GIC1 samples were prepared by mixing ASF glass ceramics with HA powder before formulated into GIC in the proportion of 3 based on the ratio as stated previously.

### 2.5. Characterization of Control and Modified GIC Samples

The cement samples were characterized by density measurement, compressive strength test, X-ray Diffraction (XRD), Fourier Transform Infrared (FTIR), Field Emission Scanning Electron Microscopy (FESEM) and Energy Dispersive X-ray (EDX).

#### 2.5.1. Density

The density (ρ) was evaluated based on Archimedes’ principle at room temperature [18]. The total of 24 samples in pellet form were weighed before being immersed into distilled water with a density of 1.0 g/cm^3^ and the test were repeated for three times. After that, the weights of each pellet were taken and resulting density with unit gram per centimeter cubic (g/cm^3^) were calculated based on formula:(1)$$\rho_{\mathrm{sample}}\;=\;\frac{W_{\mathrm{air}}}{W_{\mathrm{air}}\;-{\;W}_{\mathrm{distilled}\;\mathrm{water}}}\;\times \;\rho_{\mathrm{distilled}\;\mathrm{water}}$$
where $W_{\mathrm{air}}\;$= weight in air, $W_{\mathrm{distilled}\;\mathrm{water}}\;$= weight in distilled water and $\rho_{\mathrm{distilled}\;\mathrm{water}}\;$= density of distilled water = 1.0 g/cm^3^.

#### 2.5.2. Compressive Strength Test

Compressive strength of the samples was determined by using Universal Testing Machine (UTM) from Instron (Instron, Norwood, MA, USA) with a model of 5566. The load frame applied was up to 10 kN with crosshead of 1 mm/min and the tests were repeated up to three times to avoid error. The total of 24 samples were prepared in bulk form with cylindrical shape of 6 mm diameter and 10 mm height (6 mm × 4 mm). The compressive strength (σ) of the sample was calculated by referring to the formula:(2)$$\sigma \;=\;4F/{\pi d}^{2}$$
where, F = maximum load applied in kN, and d = the average diameter of sample in mm. The results of compressive strength were presented in MPa unit with two decimal places.

#### 2.5.3. X-ray Diffraction (XRD)

Samples in the form of powder were sent for XRD spectroscopy by using a machine of Philips (Model: PW 3040/60) with Cu K_α_ radiation equipped with 40 kV accelerating voltage and 30 mA input current. The range of 2θ value was between 20° to 80° and the outcome was examined by using X’Pert Highscore (Malvern Panalytical, Worcestershire, UK).

#### 2.5.4. Fourier Transform Infrared (FTIR) Spectroscopy

The powder samples were examined under Perkin Elmer (Model: Spectrum100 series, Perkin Elmer, Shelton, WA, USA) with wavenumber of 400 to 4000 cm^−1^ and resolution of 4 cm^−1^ for determining chemical bonding existed in the samples.

#### 2.5.5. Field Emission Scanning Electron Microscopy (FESEM)

FESEM characterization was conducted to determine morphology of the samples by using a machine from NOVA (Model: NANOSEM 230, FEI Company, Hillsboro, OR, USA) FEI brand. In this characterization, the powder samples were sputter-coated with silver coating before tested by using magnification of ×10,000.

#### 2.5.6. Energy Dispersive X-ray (EDX)

EDX testing was used for the assessment of component composition in samples. The cement samples were prepared in powder form and characterized under FEI NOVA NanoSEM 230 (FEI Company, Hillsboro, OR, USA). The EDX results were analyzed into the weight percentage of elements with two decimal places.

#### 2.5.7. Statistical Analysis

The statistical analysis used for calculation of mean and standard deviation (SD) of density and compressive strength of cement samples was one-way ANOVA model from Excel 2016. The significance level was set at 0.05. Further statistical analysis by using Tukey-Kramer post hoc test was performed to study the statistical significant difference between the samples.

## 3. Results

### 3.1. Density Analysis

Figure 1 shows the density results of GIC and GIC1 samples at different ageing time while Table 1 depicts the statistical data of the density results. According to the figure and table, the density of the GIC sample at 1 day of ageing time was 1.681 g/cm^3^. The addition of 1 wt.% of HA powder into GIC formulation represented by the GIC1 sample produced density of 1.693 g/cm^3^ at 1 day of ageing time. The density of cement samples increased with the addition of HA powder from 1.681 to 1.693 g/cm^3^. On the other hand, the density results increased as ageing time increased. This trend can be observed in both GIC and GIC1 samples. GIC sample showed the increment in density from 1.681 g/cm^3^ at 1 day of ageing time to 1.824 g/cm^3^ at 21 days of ageing time. Meanwhile, GIC1 sample showed an increase in density value from 1 day of ageing time with 1.693 g/cm^3^ to 21 day of ageing time with 1.839 g/cm^3^. The highest density results were 1.824 and 1.839 g/cm^3^ at 21 days of ageing time for GIC and GIC1 samples, respectively.

### 3.2. Compressive Strength Analysis

Figure 2 presents the compressive strength results of cement samples at different ageing time while Table 2 depicts the statistical data of the compressive strength results. The compressive strength results of GIC and GIC1 samples at 1 day of ageing time were 22.68 and 29.47 MPa, respectively. HA-added GIC was observed to result in higher compressive strength compared to GIC without HA addition, which showed increment from 22.68 to 29.47 MPa. Besides, the compressive strength results increased when ageing time increased for both samples. The compressive strength of GIC sample increased from 22.68 MPa at 1 day of ageing time to 62.78 MPa at 21 day of ageing time, while the compressive strength of GIC1 sample increased from 29.47 MPa at 1 day of ageing time to 54.34 MPa at 21 day of ageing time. The highest compressive strength results were recorded at 21 day of ageing time with 62.78 and 54.34 MPa for the respective GIC and GIC1 samples.

### 3.3. X-ray Diffraction (XRD) Analysis

Figure 3 shows the XRD patterns of GIC and GIC1 samples at different ageing times. According to Figure 3, the formation of fluorapatite (FA; ICDD file No. 98-001-7206) crystal phase was observed in the GIC sample, from 1 day until 21 days of ageing time. For GIC1 sample at different ageing times, the addition of HA powder into GIC resulted in the detection of fluorohydroxyapatite (FHA; ICDD file No. 98-008-0180) crystal phase, as seen in Figure 3. Based on the observation from Figure 3, the highest crystal peak was detected at the same diffraction angle, which was 2θ = 32°. No significant difference of XRD pattern was observed from 1, 7, 14 and 21 days of ageing time for both GIC and GIC1 samples.

### 3.4. Fourier Transform Infrared (FTIR) Analysis

Figure 4 shows the FTIR patterns at different ageing times for the GIC and GIC1 samples. Meanwhile, Table 3 depicts the vibrational modes assigned for FTIR spectra that existed in the cement samples. By referring to Figure 4, v_2_ O–P–O bending vibrational mode was detected at ~440 cm^−1^ while v_4_ O–P–O bending mode was discovered at ~570 and ~600 cm^−1^. Besides, the existence of the FTIR spectral band at ~1020 cm^−1^ represented the vibrational mode of v_3_ asymmetric P–O stretching. The detection of C–O and asymmetric COOH vibration modes were found at wavenumbers ~1460 and ~1550 cm^−1^, respectively. A broad and wide peak was discovered at ~3400 cm^−1^ indicated OH vibration mode. The FTIR pattern of GIC1 sample at different ageing time revealed the similar vibrational modes existed in the sample when compared to GIC sample. However, the only difference found was the existence of OH–F vibration mode at ~3550 cm^−1^.

### 3.5. Field Emission Scanning Electron Microscopy (FESEM) Analysis

Microstructures of GIC and GIC1 samples at different ageing times are shown in Figure 5. The observation of irregular shape of glass ceramics structure with non-uniform particle distribution was detected in both cement samples, as seen in FESEM images. The formation of needle-like and spherical particles was observed on the surface of glass ceramics particles. Besides, no significant difference was detected in the samples when compared to samples with different ageing time.

### 3.6. Energy Dispersive X-ray (EDX) Analysis

Table 4 shows the chemical composition of the GIC and GIC1 samples at 1 day of ageing time. The chemical elements existed were oxygen (O), carbon (C), calcium (Ca), aluminum (Al), phosphorus (P), silicon (Si), fluorine (F) and sodium (Na). The most element detected in GIC and GIC1 samples was O with the respective 42.54 and 45.20%. Meanwhile, C existed with 22.86 and 20.90%, while Al element was found in the GIC and GIC1 sample with 7.18 and 7.11%, respectively. The detection of P with 5.11 and 5.16% as well as Si element with 4.33 and 3.67% in the respective GIC and GIC1 were recorded from EDX analysis. Element F existed as a minor element with 3.87 and 3.91% while Na consisted at about 1.30 and 1.15% in the respective GIC and GIC1 samples.

## 4. Discussion

In this work, density measurement was used to observe the physical properties of both GIC and HA-added GIC samples at different ageing time. The density results in Figure 1 and Table 1 revealed an increment in density of GIC with the addition of HA powder. This is due to high density of HA compared to ASF glass ceramics samples. In this case, the measured density of ASF glass ceramics was 2.561 g/cm^3^ while the density of HA used in this work was 2.673 g/cm^3^. According to a study, the increase in the density of HA-added GIC samples agrees qualitatively with the one, as predicted by the composition relation, and might be due to the replacement of high density of HA powders [20]. Thus, the density of HA which has a higher density was proposed to dominate the density of cement samples. Besides, the density of GIC and GIC1 samples was observed to increase as the ageing time increased from 1 day to 21 days. The increase in density along with ageing time is due to the increment in ratio of bound to unbound water [25,28]. The reaction in the GIC matrix occurs with the presence of water as a medium of setting reaction over a period of time. Moreover, water acts as a medium of reaction which helps in maturation and hardening process of cement.

The results of compressive strength test presented in Figure 2 and Table 2 found that the addition of HA into GIC resulted in higher compressive strength compared to GIC without HA addition. This is due to the presence of crystal phase in the glass structure of glass ionomer [23,29]. Metal ions such as Ca^2+^ are displaced from the added HA, therefore increased the participation of ions in acid base reaction. The setting reaction that occurs when there is addition of HA into GIC has been explained in few studies [21,30]. Upon addition of HA into GIC, H^+^ from acid polymer attacks the ceramic particles in the polysalt bridge formation and cross-linking, thus forming an intermediate layer from the interaction. The intermediate layer is very resistant to acid and is difficult to break. Therefore, the addition of HA into GIC enhance the mechanical strength of the resulting GIC [23,30,31]. Figure 6 explained the schematic diagram of setting reaction of HA added GIC.

Besides, the compressive strength of both GIC and GIC1 samples increased as ageing time increased. The immersion of cements into deionized water as a medium of reaction at different periods of time allows reaction between the GIC matrix to occur. Longer ageing time seemed to improve the strength of the GIC. Compressive strength results of the cement samples are related to the density of the samples. The density of GIC samples affects the mechanical properties of the set cement [4]. The addition of HA resulted in higher density compared to GIC without the HA addition. Such results are in line with the observations of Goenka et al. [4] who reported that the samples of the HA-added GIC resulted in increased hardness due to increased density of the set cement. Besides, some studies reported that the addition of certain volume of HA and the role of PAA in cement caused an increase in crosslink density of the cement structure, thus helping to increase the mechanical strength of the GIC [4,28].

Through Turkey-Kramer post-hoc evaluation, the significant difference in density and compressive strength of each cement samples when compared between different ageing times was determined and the results were presented in Table 1 and Table 2, respectively. According to density results in Table 1, the comparison between different ageing time revealed that not all GIC and GIC1 samples showed significantly different results. Meanwhile, the compressive strength results shown in Table 2 revealed the significantly different results when compared to different ageing time and this situation occurred in both GIC and GIC1 samples. The increase in ageing time caused the significant increase in compressive strength value. Several studies reported that no significant difference was observed in compressive strength between GIC without HA addition and HA-added GIC samples at different ageing time [32,33]. Thereby, it can be concluded that HA addition into GIC composition did not give any significant effect in compressive strength of the resulting GIC.

XRD patterns in Figure 3 revealed the formation of FA crystal phase in GIC samples at 1, 7, 14 and 21 days of ageing time. This is due to the utilization of ASF glass ceramics as base powder in the formulation of GIC. The study of ASF glass ceramics had been done by some researchers and found that the ASF glass ceramics with similar composition caused the development of FA crystal phase [10,11,12]. The addition of a small weight percentage of HA did not affect the XRD pattern except for the detection of FHA phase. According to some studies, the formation of FHA in modified GIC samples of different ageing time is due to the substitution of hydroxyl ion (OH^−^) by fluoride ion (F^−^) that occurred in the apatite composition [28,34]. The reaction of the released fluoride ion with the HA crystal was summarized in general by Equation (3) [35]. In this case, the substitution of F^‒^ ion into the apatite crystal caused the formation of the FHA crystal phase. FHA crystal was found to occur naturally in human bone and teeth. Owing better mechanical and chemical properties, FA and FHA have been attracting a great attention in biomedical application especially for dental restoration [36].
(3)$${\mathrm{Ca}}_{10}{{(\mathrm{PO}}_{4})}_{6}{\left(\mathrm{OH}\right)}_{2}{\;+\;F}^{-}\to {\mathrm{Ca}}_{10}{{(\mathrm{PO}}_{4})}_{6}{\mathrm{FOH}\;+\;\mathrm{OH}}^{-}$$

FTIR patterns of GIC and GIC1 samples at different ageing times depicted in Figure 4 revealed the existence of similar spectral bands. The appearance of the spectral band at ~400 cm^−1^ was belonged to double degenerate O–P–O bending mode (v_2_) indicated by phosphate group in apatite sample [19,20]. Next, the emergence of triple degenerate O–P–O bending mode (v_4_) which appeared at both intense peaks at wavenumber ~570 and ~600 cm^−1^. Another occurrence of phosphate chemical bonding was observed at an intense peak of ~1020 cm^−1^ which represented asymmetric P–O stretching mode (v_3_). According to related literature, more than one distinction site of PO_4_^3−^ vibration modes was found in the FTIR spectral wavenumber of cement sample which proved the formation of apatite phase inside the cement samples [4,21,22,23,24].

On the other hand, the carbonate group depicted by C–O vibration mode at wavenumber ~1460 cm^−1^ confirmed the involvement of carbonate precursors in the production of GIC samples. Besides, Khiri and co-workers explained the formation of asymmetric COOH band at 1550 cm^−1^ which originated from the carboxyl group in PAA used in the formulation of GIC [25]. The cross-linking reaction of the carboxylic group was observed in all cement samples due to an active reaction that occurred between PAA and ASF glass ceramics. Next, a broad absorption band was discovered at high frequency of wavenumber ~3400 cm^−1^ which indicated the occurrence of OH vibration mode. Jekonovic et al. [26] and Montazeri et al. [27] stated that the occurrence of the vibrational mode is due to stretching of intermolecular H bonds caused by water absorption. At the same time, the existence of this spectral band showed the presence of water during cement preparation [26].

The only difference spotted in FTIR spectral of GIC1 sample compared to GIC sample was the existence of hydroxyl group by incorporation of fluorine ion which was represented by OH–F vibration mode at wavenumber ~3550 cm^−1^, claimed the formation of FHA phase in GIC1 sample. Studies from Eslamia et al. [37] and Jokanović et al. [27] explained that the band existed at 3550 cm^−1^ in HA samples due to stretching mode of hydrogen bonded OH^−^ ion. When hydroxyl groups are partially replaced with fluoride ions in HA, the stretching mode of OH^−^ can shift to the new band that arises from OH-F bond. In FHA, OH band appeared at around 3550 cm^−1^ indicating that the OH–F interaction influences all the OH^−^ ions present in the samples.

FESEM morphology of both GIC and GIC1 samples in Figure 5 showed the irregular shape of glass ceramics structure with non-uniform particle distribution. The detection of apatite crystals by the formation of needle-like and spherical particles were observed to disperse on the surface of the glass ceramics particle. This observation is in conjunction with the findings found by Khaghani and co-workers, who investigated the SEM structure of HA incorporation into GIC composition [29]. According to some literatures, hexagonal structure of apatite was illustrated by rod or needle-like and spherical structure [26,27,38]. Due to high adsorbability, apatite crystals with rod-like morphology have shown favorable biocompatibility and bioactivity, as the corresponding interactions of Van der Waals are proportional to the large rod surface region [39,40].

EDX analysis was utilized to investigate the chemical composition of cement samples at 1 day of ageing time and the results were presented in Table 4. Most of the elements detected by EDX test originated from chemical compounds used in the preparation of glass ceramics powder which then used as base powder for GIC formulation, meanwhile C was developed from polyacid used in initiating the reaction of GIC. Oxygen existed as the major element consisted in the cement samples due to consumption of oxide elements in the fabrication of ASF glass ceramics powder. A study from Rahman et al. [12] explained the existence of minor elements such as Na, Si, Al, and other elements make up a majority of SLS glasses, which then utilized in the making of base powder of the cement. In tooth structure, the hardest and highly mineralized substance, called enamel consists of 96% inorganic matter, 1–2% organic matter and 2–3% water. The majority of inorganic matter which make up the enamel structure is owned by distinct crystalline phases of HA. Calcium (Ca) and phosphorus (P) are essential elements composed in HA, and the composition of both elements is crucial for the saturation of apatite that improves the process of remineralization [10].

## 5. Conclusions

In the current study, CS and SLS glass had been exploited to fabricate ASF glass ceramics which subsequently used in the formulation of GIC. The physical, mechanical and structural properties of GIC formulated based on ASF glass ceramics as base silicate powder was enhanced by the addition of HA powder, which revealed the enhancement in density and compressive strength test along with the formation of FHA crystal phase from structural studies. Besides, ageing time also had been observed to improve the properties of the resulting GIC. In a nutshell, the improved properties of physical, structural and mechanical encourage the use of HA-added GIC in biomedical application especially for dental restoration.

## Figures and Tables

**Figure 1 materials-14-00954-f001:**
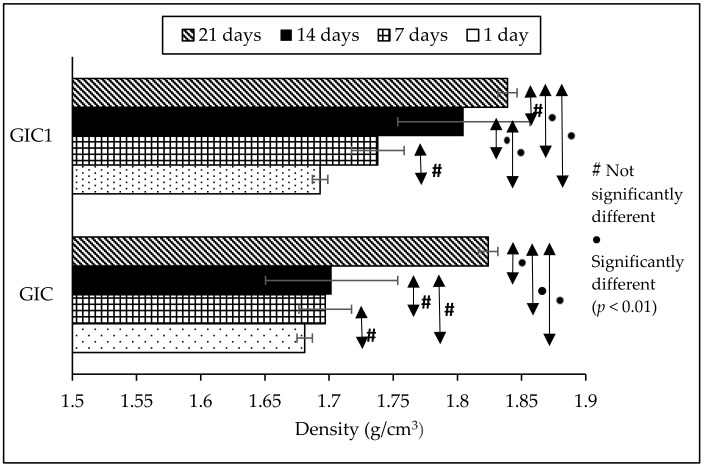
Density results of GIC and GIC1 samples at different ageing time (error bars indicate standard error).

**Figure 2 materials-14-00954-f002:**
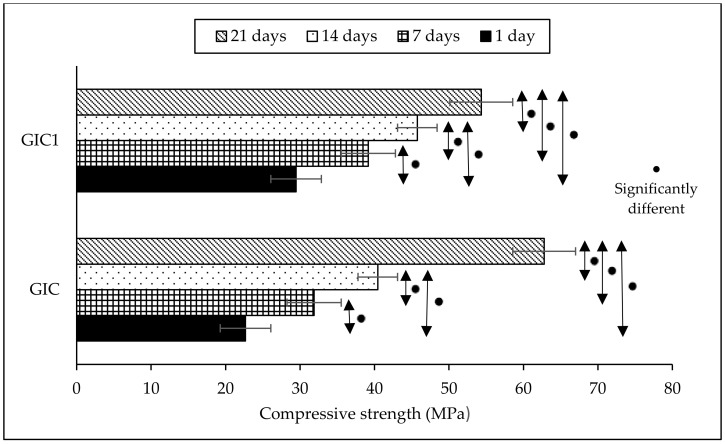
Compressive strength results of GIC and GIC1 samples at different ageing time (error bars indicate standard error).

**Figure 3 materials-14-00954-f003:**
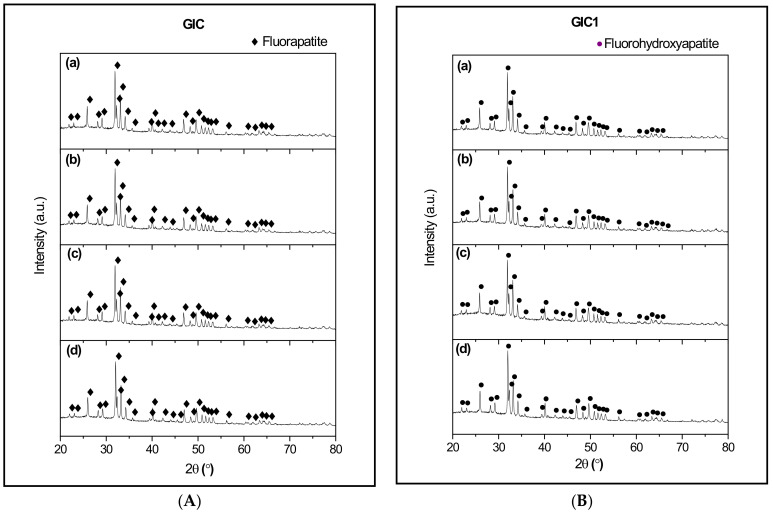
XRD patterns of GIC (**A**) and GIC1 (**B**) sample with (a) 1 day, (b) 7 days, (c) 14 days, and (d) 21 days of ageing time.

**Figure 4 materials-14-00954-f004:**
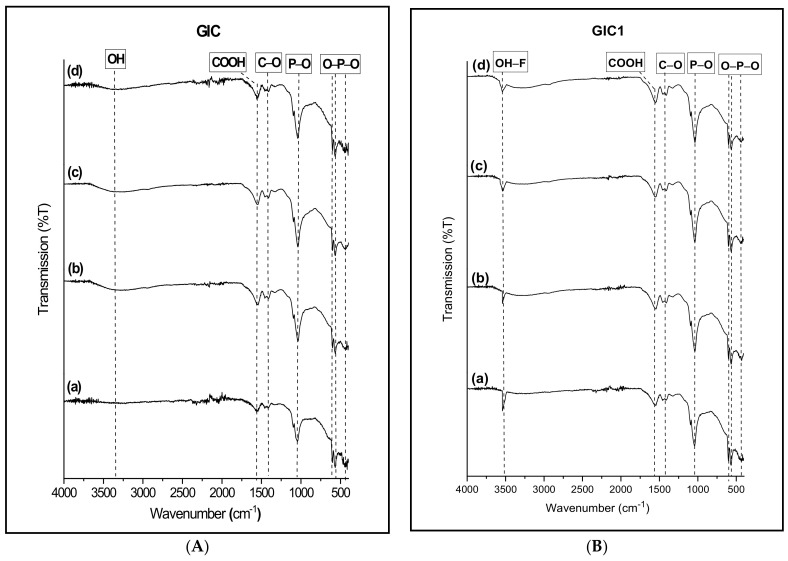
FTIR pattern of GIC (**A**) and GIC1 (**B**) sample for (a) 1 day, (b) 7 days, (c) 14 days, and (d) 21 days of ageing time.

**Figure 5 materials-14-00954-f005:**
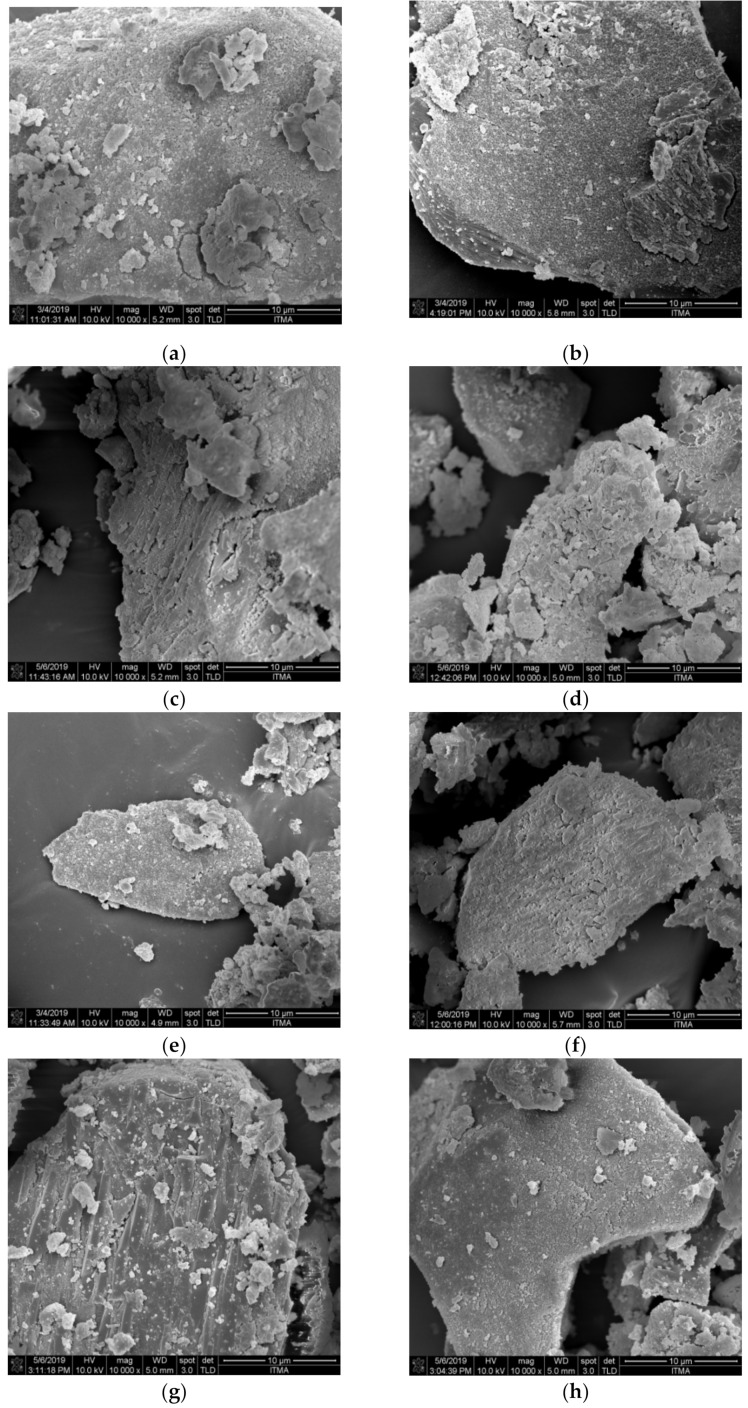
FESEM micrograph under magnification of 10,000× for samples (**a**) GIC 1 day, (**b**) GIC 7 days, (**c**) GIC 14 days, (**d**) GIC 21 days, (**e**) GIC1 1 day, (**f**) GIC1 7 days, (**g**) GIC1 14 days, and (**h**) GIC1 21 days.

**Figure 6 materials-14-00954-f006:**
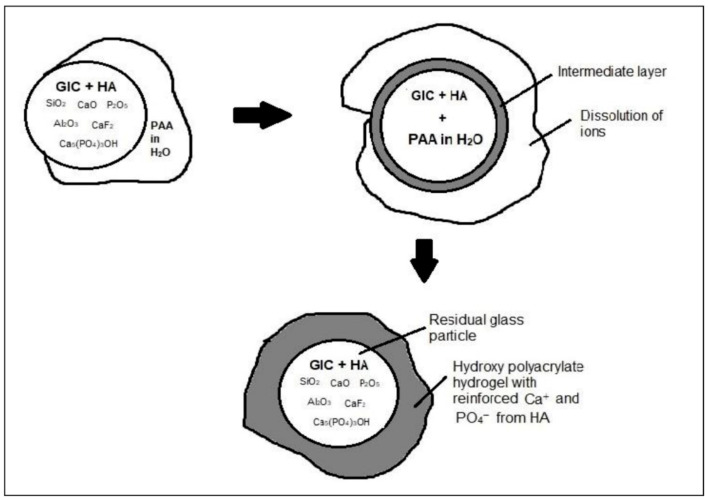
Schematic diagram of setting reaction of HA-added GIC.

**Table 1 materials-14-00954-t001:** Mean density along with the standard deviation (SD) of the GIC and GIC1 samples at different ageing time.

Density of Sample (g/cm^3^)	Ageing Time			
	1 Day	7 Days	14 Days	21 Days
**GIC**	1.681 (0.011) ^a^	1.697 (0.006) ^a^	1.702 (0.015) ^a^	1.824 (0.043) ^b^
**GIC1**	1.693 (0.015) ^a^	1.738 (0.033) ^a^	1.805 (0.007) ^b^	1.839 (0.016) ^b^

The different superscript letters (example: ^a^ and ^b^) for each sample (each row) represent significant difference between ageing time (*p* < 0.01). Meanwhile, similar superscript letters (example: ^a^ and ^a^) represent no significance difference between ageing time.

**Table 2 materials-14-00954-t002:** Mean compressive strength along with standard deviation (SD) of GIC and GIC1 samples at different ageing time.

Compressive Strength of Sample (MPa)	Ageing Time			
	1 Day	7 Days	14 Days	21 Days
**GIC**	22.68 (0.73) ^a^	31.89 (0.86) ^b^	40.44 (0.58) ^c^	62.78 (0.21) ^d^
**GIC1**	29.47 (2.17) ^a^	39.17 (1.24) ^b^	45.75 (5.11) ^c^	54.34 (0.63) ^d^

The different superscript letters (example: ^a^ and ^b^) for each sample (each row) represent significant difference between ageing time (*p* < 0.01). Meanwhile, similar superscript letters (example: ^a^ and ^a^) represent no significance difference between ageing time.

**Table 3 materials-14-00954-t003:** Vibrational modes assigned for GIC and GIC1 samples.

Wavenumber (cm^−1^)	Vibrational Mode	References
~440	v_2_ O–P–O bending	[19,20]
~570, ~600	v_4_ O–P–O bending	[4,21,22,23,24]
~1020	v_3_ asymmetric P‒O stretching	[19,20]
~1460	C–O vibration	[21]
~1550	Asymmetric COOH	[25]
~3400	OH vibration	[26,27]
~3550	OH–F vibration	[26,27]

**Table 4 materials-14-00954-t004:** Chemical composition of GIC and GIC1 samples at 1 day of ageing time.

Element	Weight Percentage (%)	
	GIC	GIC1
O	42.54	45.20
C	22.86	20.90
Ca	12.81	12.90
Al	7.18	7.11
P	5.11	5.16
Si	4.33	3.67
F	3.87	3.91
Na	1.30	1.15
Total	100.00	100.00

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
