# Peer review of "Incorporation of Hydroxyapatite into Glass Ionomer Cement (GIC) Formulated Based on Alumino-Silicate-Fluoride Glass Ceramics from Waste Materials"

_materials, 2021, doi:10.3390/ma14040954_

Round 1

Reviewer 1 Report

1. The paper has many basic language errors. The paper needs expert editing by a native English speaker, and use of a professional editing service is stroingly advised.

2. The last part of the introduction needs to explain clearly how this work differs from past studies - as currently written it is too vague and imprecise.

3. The source of all key chemical reagents must be stated. This includes the clam shells (CS) and the soda lime silicate (SLS) glass, and the polyacrylic acid.

4. The calcination time needs to be stated.

5. The method of grinding the frits needs to be stated.

6. The head movement speed for compressive strngty testing needs to be stated.

7. The number of samples prepared for testing (N) (e.g. density, compressive strength) needs to be stated. Information on data distribution (normal or not) and statistical comparisons is lacking and needs to be included. Where appropriate, how does the regular GIC compare with the HA supplemented GIC? Is it actually stronger? Comments such as " The addition of
252 HA resulted in higher density compared to GIC without the HA addition." need to be supported with statistics.

8. Legends to graphs (Figs 2 and 3) need to state what the error bars are (e.g. standard deviation?).

9. The paper needs reorganizing with images incorporated into the flow of the text as per the instructions to authors. The layout of the figures 4 and onwards should combine together GIC and GIC1 side by side for ease of comparison.

10. Did the authors assess water imbibition and GIC sample weight loss/weight gain over the 21 day period? if not, why not?

11. Why is the chemical composition of the HA-enriched GIC missing from table 2?

12. The discussion does not explicitly state how the HA-enriched GIC compared to the normal GIC, e.g. compressive strngth head-to-head at each of the timepoints.

13. The authors should make reference to existing commercial GIC materials in terms of how their two experimental materials performed - e.g. compressive strength values. Were these typical or not for chemically cured conventional GIC materials?

14. The last part of the discussion is illogical. The study had widely ranging Ca/P ratios.
There was no assessment of biocompatibility. Hence it is NOT possible to say "Thus, it can be concluded that the GIC and HA-added GIC are suitable for implantation purposes".

15. Only 25 references are given - but the text has references up to 46.

16. The references need to be done to the correct style - use of bold and italics.

Reviewer 2 Report

Thank you for the opportunity to review the manuscript.

The aim of the manuscript was to study the effect of hydroxyapatite (HA) addition into GIC (ASF-based) based on physical, mechanical and structural properties. The authors concluded that HA in GIC will improve the properties of the cement. The study contributors had used several characterization techniques to study the HA incorporated GIC properties including XRD, FTIR, FE-SEM, and EDX. The study appears comprehensively performed, however, would need major revision before it can be further scrutinized to assess its suitability for publication:

Abstract:

  1. The authors have used abbreviations in the text; while the abbreviations were further not mentioned in the abstract. As a usual research writing practice, the abbreviations are used when its mention would subsequently appear in the written text.
  2. Line 23-24: Please mention the concentration of HA in GIC.
  3. Line 25: The authors mention – “the addition of HA caused an increment….”. Is it that authors want to write that addition of HA caused an increase in density and compressive strength?
  4. Line 32: “The involvement of HA….”. Is it the incorporation of HA?

Introduction:

  1. The introduction is too long to read and takes time to reach the research gap. Also, the research gap in very vaguely mentioned in the last para of the introduction. Needs revision on these major points. More emphasis should be on ASF glass-ceramics rather than the history of GIC as known by all in the fraternity.
  2. Line 51: Kindly specify what do you mean by anti-cariogenic?
  3. Line 82: Kindly move the sentence on Figure 1 to the discussion section.

Materials and methods:

  1. Throughout the materials and methods, the authors need to specify the material used along with a batch number (if any) and information on manufacturer/company, brand, city, country needs to be mentioned in the brackets.
  2. Line 130 – Can the authors justify the use of only one concentration of HA?
  3. Explain - why a commercial GIC control was not used to compare the properties of ASF containing GIC and HA+ASF containing GIC?
  4. Appropriate references in the methodology used are missing. Kindly include them.
  5. The section needs to be written in the past tense. See line 146 as an example to rectify.
  6. Line 164 – How was the field for scanning chosen?
  7. Please add a note on the statistical analysis used in this study.

Results:

  1. The results section needs to be revised with appropriate tenses. Kindly see the comment on tenses in the materials and methods section.
  2. The density results increases…” - how was this compared? Where are the details of the statistical test used?
  3. HA-added GIC was observed to result in higher compressive strength compared to GIC without HA addition” – how was this compared? What statistical test was used? The entire results section need revision. Kindly describe the results on the basis of the appropriate statistical tests used.  
  4. For ease of comparison, can Figure 4 and Figure 5 be combined?
  5. Line 194 – it is mentioned that “no significant difference of XRD pattern was observed….”How was this tested?
  6. Similar to comment 4, can Figure 6 and Figure 7 be combined?
  7. Also, combine Figure 8 and 9 to identify the differences in one figure.
  8. Table 2 and 3 – Kindly present mean and SD values including the statistical tests used to identify the differences between the group.
  9. Line 215 to 224 – Whether the Ca/P ratio significantly increased or decreased? Statistical significance? Co-efficient of variation? It needs a statistical comparison.

Discussion:

  1. The main results of the study, tested the null/alternate hypothesis, results of the test and novel contributions are missing in the opening para of the discussion.
  2. Most of the discussion talks about the interaction mechanism. The discussion section needs to be revised once the results have been revised.

Conclusion:

The conclusion is a mere repetition of the results as opposed to answering the aim of the study. Kindly revise.

Overall, the authors of the study have invested efforts; however, technical aspects of reporting the work, analysing the data, data interpretation and presentation are missing from the manuscript. Needs thorough and major revision.

Round 2

Reviewer 1 Report

The paper has been greatly improved by the revisions that have been made.

Author Response

The English language has been revised for several times to avoid error.

Reviewer 2 Report

Dear authors, 

Thank you for revising the manuscript. The revised version of the manuscript is very well improved. Taking further, a major revision is still required: 

As mentioned by the authors' reg. the statistical tests, only mean and SD were used to present the data. When the authors say that they have used 1-way ANOVA model for statistical analysis, the test can only define whether there is a difference between the groups or not. Further post-hoc tests are needed to delineate, how and where the difference is between the groups and its magnitude based on p-values. Please make this revision and certainly subsequent revision will follow in the presentation of the results with discussion section of the manuscript. The authors should carefully review and revise this important section before any further comments can be made on the manuscript. Please make sure you address this point so minimum reviews are required before a final decision from the reviewer can be made. 

Thank you once again. Keep up the good work!

Author Response

1. English language and style

  • The English language has been revised and checked for several times to reduce errors.

2. As mentioned by the authors' reg. the statistical tests, only mean and SD were used to present the data. When the authors say that they have used 1-way ANOVA model for statistical analysis, the test can only define whether there is a difference between the groups or not. Further post-hoc tests are needed to delineate, how and where the difference is between the groups and its magnitude based on p-values. Please make this revision and certainly subsequent revision will follow in the presentation of the results with discussion section of the manuscript. The authors should carefully review and revise this important section before any further comments can be made on the manuscript. Please make sure you address this point so minimum reviews are required before a final decision from the reviewer can be made.

  • As suggested by reviewer, the post-hoc test has been studied and the results has been presented in Table 1 (Section 3.1 and Line 189) and Table 2 (Section 3.2 and Line 207). The results of the post-hoc test also had been discussed in the discussion part (Section 4, Line 299 to 309). 

Round 3

Reviewer 2 Report

The manuscript is fine to be accepted provided a minor change is required. Prior to final acceptance, the authors should also do the posthoc tests for Figure 1 and 2.

Author Response

1. English language and style are fine/minor spell check required

  • English language has been checked and several times to reduce errors.

2. The authors should also do the posthoc tests for Figure 1 and 2.

  • The results of posthoc tests has been presented in Figures 1 and 2 (Section 3.1 and 3.2) as suggested by reviewer.